# The Reproductive Biology of Puye (*Galaxias maculatus*) under Experimental Culture

**DOI:** 10.3390/ani14020320

**Published:** 2024-01-19

**Authors:** Leydy Sandoval-Vargas, Maritza Pérez-Atehortúa, Elías Figueroa Villalobos, José Zamorano, Iván Valdebenito

**Affiliations:** 1Faculty of Natural Resources, Catholic University of Temuco, Temuco 4813302, Chile; leydyasminsandoval@gmail.com (L.S.-V.); maritza.perez2021@alu.uct.cl (M.P.-A.); 2Nucleus of Research in Food Production, Faculty of Natural Resources, Catholic University of Temuco, Temuco 4813302, Chile; efigueroa@uct.cl; 3Faro Verde, Company, Temuco 4811102, Chile; jzamorano@faroverde.cl

**Keywords:** fecundity, reproduction, sexual maturity, spawning period

## Abstract

**Simple Summary:**

*Galaxias maculatus* plays an important ecological and socioeconomic role in Chile. We determine the sex ratio, sexual maturity stages, gonadosomatic index, spawning period, type and frequency of spawning, and fecundity under experimental culture. The sex ratio was close to 1:1. First sexual maturity was reached at one year of age in 50% of the population. The highest GSI in both females and males was found in August. Nevertheless, females have a long spawning period, divided into two subperiods; the highest reproductive peak occurs between September and October, and the minor peak occurs between December and February. The females spawned between 3 and 10 times over a period of two months. The number of embryos per female per day varied from 1 to 429, while the total number of embryos per female during the entire season varied from 163 to 1044. This knowledge will be useful to establish future reproductive programs in captivity as a strategy for sustainable fishery and aquaculture management.

**Abstract:**

This study determines the reproductive patterns of puye (*Galaxias maculatus*) under culture conditions. A population of 567 wild fish was caught in the Cautín River, Chile, and held in captivity for four years. Mortality, sex ratio, gonadosomatic index (GSI), sexual maturity stages, spawning period, type and frequency of spawning, and fecundity were measured. The fish grew throughout the experimental period, with the fastest rate during the first half of the first year of life. The highest mortality occurred during the first three months of the experiment and during the spawning season. The sex ratio was almost 1:1 (female:male). First sexual maturity was reached at one year of age, with an average weight of 0.85 ± 0.01 g, total length of 4.85 ± 0.16 cm, and condition factor 0.0074. The highest GSI in both females (12.14 ± 0.74) and males (17.7 ± 2.70) was recorded in August. Nevertheless, the females spawned 3 to 10 times between September and February, with the highest reproductive peak between September and October. The number of embryos per female per day varied from 1 to 429, while the total number of embryos per female during the entire season evaluated varied from 163 to 1044. There was a high correlation (r = 0.82) between absolute fecundity and body weight. Although further studies are needed in this field, these results are basic for establishing future reproductive programs in captivity as a strategy for sustainable fisheries and aquaculture management.

## 1. Introduction

The puye, *Galaxias maculatus* (Jenyns, 1842), is a small fish with diadromous and landlocked populations distributed throughout circum-Antarctic countries [1,2,3]. In South America, the species is distributed throughout southern Argentina and Chile [4], where it is characterised as freshwater or marginally catadromous [5,6].

*G. maculatus* has socioeconomic and ecological importance in some regions, especially rural areas where it is captured and consumed [7]. In recent decades, populations and catch volumes in Chile have experienced a marked variation due to overfishing and predation, mainly by introduced salmonids. At the end of the 1960s, the catch volume of this species in Chile varied between 1.5 and 4 tonnes per year; after 1990, its fishing was expanded to other regions of the country, reaching thus 14 tonnes per year [8]. Nevertheless, the subsequent high pressure exerted by artisanal fishing, together with predation and the disturbances of habitat, led to the species being classified as vulnerable in the 1990s, making necessary the protection of the species by the Aquaculture and Fisheries Agency [9]. Although the populations have been recently recovered, and a catch volume of three tonnes was reported in 2022 [10], it is important to study the reproductive cycle captivity to promote commercial sustainability. 

The high commercial value of the species is due to its similarity to the crystalline stage of the European eel. The body is transparent in the juvenile stage, and the fish is caught and marketed as a culinary substitute for the crystalline eel [11]. In ecological terms, *G. maculatus* is key to the functioning of the region’s aquatic systems. It is the most abundant forage fish consumed by other fish species, and thus reduces the pressure on other native species [12]. The small size of *G. maculatus,* together with its great life history plasticity, makes the species a good model fish for both laboratory and field research. Some biological aspects, such as habitat [6,13,14], reproductive cycle in the natural habitat [2,3,15], and potential for culture [8,16,17], have been well studied. Responses to parasites [18] and toxic substances [16,19] have also been researched. 

In reproductive terms, *G. maculatus* displays a specialised reproductive strategy, which consists of spawning on riparian vegetation that is inundated during winter and by high spring tides [20]. The adhesive eggs remain stuck to the lower part of terrestrial vegetation and aerial roots until the water level rises to transport the hatched larvae to the river, lake, or sea [3,21]. In New Zealand, gonad maturation begins in the summer, with the subsequent spawning season in autumn and early winter [5]. The spawning period specifically occurs from March to May, with peak movement between late April and mid-May [22]. In South America, particularly in Tierra del Fuego, Argentina, the reproductive season extends from October to February. 

Nevertheless, most of the efforts to assess the reproductive biology of this species have been carried out in wild conditions [15], but their reproductive parameters in captivity have not been described in detail. Understanding the life cycle and reproductive parameters such as longevity, spawning season, type and frequency of spawning, and fecundity (number of oocytes produced) are crucial to determining the reproductive potential of an species with commercial proposes [23]. Given the ongoing interest in this species due to their high commercial value, this information will be useful to establish reproductive management in aquaculture, contributing thus to sustainable fishery. 

On the other hand, for several years, there has been a great controversy regarding if this species is semelparous or not [24,25]. McDowall et al. [24] classifies *G. maculatus* as predominantly a semelparous, annual species, while Stevens et al. [25] demonstrated using histological studies that some individuals can survive post-spawning, but the percentage of survival is unknown. In addition, it has been described that the final sexual maturation of this species is not reached until the fish migrate short distances upstream [5]. In this context, we hypothesised that *G. maculatus* was not able to mature sexually or spawn spontaneously in captivity. Secondary, we also hypothesised that most of the individuals would die after the first reproductive event. For these reasons, and because sustainable aquaculture practices require independence from the natural environment, the objective of this study is to identify the reproductive pattern parameters of *G. maculatus* under culture conditions in southern Chile. 

## 2. Materials and Methods

### 2.1. Fish Maintenance and Handling

Five hundred and sixty-seven wild *G. maculatus* juveniles with an average body weight of 0.12 ± 0.01 and body length of 3.06 ± 0.13 were caught in the Cautín River (30° S:72° W) using a trawl net. The fish were then transported to a research unit at the Universidad Católica de Temuco, Chile, where they were acclimatised to the experimental feeding regimen for 2 weeks before starting the experiment. Throughout the research period, the individuals were held at a stock density of 2.7 kg/m^3^ in two 100 L fiberglass tanks, with a 1.5% daily freshwater exchange and constant aeration. The fish were fed *ad libitum* four times a day, with brine shrimp artemia (*Crustacea*, *Anostraca*) as a first feed. Juveniles and adults were fed with a salmonid diet (56% crude protein, 18% crude lipids, 8.4% carbohydrates, and 12% ash; Skretting, Osorno, Chile). The dissolved oxygen, temperature (YSI, 550A, USA, EcoSense DO200A, YSI, Yellow Spring, OH, USA) and pH of the water were registered daily prior to cleaning and extraction of dead fish.

### 2.2. Fish Performance

During the four years of the experiment, twenty-five fish were sampled at monthly intervals to determine their weight (W in g) and total length (L in cm). From this information, Fulton’s condition factor (K-factor = W/L^3^ × 100) and the instantaneous growth rate or gain in weight (GW = (lnW_f_ − lnW_i_)/T, where GW = gain in weight, lnW_f_ = natural logarithm of final weight, lnW_i_ = natural logarithm of initial weight, and T = number of days between sampling) and length (GL = (lnL_f_ − lnL_i_)/T, where GL = gain in length, lnL_f_ = natural logarithm of final length, lnL_i_ = natural logarithm of initial length, and T = number of days between sampling) were estimated. Additionally, survival and lifespan were recorded. These and additional determined parameters are shown in Figure 1. 

### 2.3. Gonadosomatic Index (GSI)

To determine the GSI variation throughout the year, a group of 240 fish offspring from the second spawning of the caught population were selected. A total of 20 individuals—10 males and 10 females—whose sex could be determined macroscopically were sacrificed monthly. The GSI was calculated for each fish as GSI = [gonad weight (g)/total weight fish (g)] × 100.

### 2.4. Sexual Maturity Stages

The sexual maturity stage was estimated based on GSI index and a macroscopic analyses carried out by visual inspection according to Mardones et al. [8]. This is possible because the species shows a transparent abdomen and sexual dimorphism during the adult phase. Briefly, twenty-five fish were examined macroscopically each month. First, the fish were anaesthetised in small groups using 0.3 mL/L% of BZ-20 (benzoate salt solution at 20%). Afterwards, the abdomen and genital pore were examined carefully by the same person. For immaturity stage, it was not possible to differentiate males from females since the abdominal regions of both females and males are silver. From initial maturity to maximum maturity, the females are recognised because the eggs are visible close to genital pore, and the abdomen increases more, while the genital pore seems protruded. In males, the testicles are observed since the initial maturity to resorption; the abdomen increases also in volume, and in the maximum maturity stage, milt is easily released after gentle abdominal massage. Nevertheless, the resorption stage in males was not easy to identify because the abdomen seemed a little bulging and milt were not released. Sex and maturity stages were designated following the scale described in Appendix A (adapted from Mardones et al. [8]).

Furthermore, the sex ratio was determined in 165 fish from the first filial generation (F1) in captivity through sexual dimorphism. The maturity stages were defined as immaturity (virginal), initial maturity, advanced maturity, maximum maturity, and resorption, as proposed by Peredo and Sobarzo [4]. 

### 2.5. Spawning Period and Spawning Frequency Determination

To determine the spawning period, the culture tanks were conditioned with a natural substrate throughout the year. These materials (roots, juncus, and rocks) were extracted from the same area where the fish were caught. The tanks were visually inspected each day to identify the presence of oocytes on the substrate. The number of embryos and the date were recorded. 

To determine spawning frequency, 6 female and 12 male *G. maculatus* with signs of full maturity were selected and distributed in a 1:2 ratio (1 female:2 males) in six 30 L tanks, conditioned as above. The tanks were monitored each day to detect the presence of embryos. Each date was recorded to determine seasonality. 

### 2.6. Fecundity Estimation 

The batch fecundity (number of oocytes released female in each spawning event) was determined in 80 sexually mature females, which were weighed before oocyte extraction. The oocytes were extracted by applying slight abdominal pressure, placed individually in a Petri dish, and finally, counted under a stereomicroscope (Olympus SMZ-2T; Olympus Co., Tokyo, Japan). 

### 2.7. Statistical Analysis

Data are shown as mean ± standard deviation ($\bar{X}$ ± SD). Normality and homogeneity of the data of weight (g), length (cm), K-factor, GSI, and total number of embryos were verified by the Shapiro–Wilk and Bartlett tests, respectively. Those parameters were analysed using nonparametric T-test (Kruskal–Wallis test) with Holm adjustment. The level of significance was set at *p* < 0.05. The Spearman method was used to analyse the correlation between total length, weight, and batch fecundity. All statistical analyses were performed using R 4.2.2 (R Core Development Team). Photoperiod frequencies, temperature, GSI, K-factor, and gain were plotted using the SigmaPlot Version 15 programme. 

## 3. Results

Figure 2A shows the temperature and photoperiod over the four years of evaluation. The temperature ranged between 6.9 °C (lowest temperature in winter) and 18.6 °C (highest temperature in summer). A natural photoperiod regime of 9 and 15 h light was registered in winter and summer, respectively. Dissolved oxygen and pH in the water was 8.3 ± 1.12 mg/L. pH 6.85 ± 0.52, respectively. 

### 3.1. Fish Performance

The survival rate (Figure 2B) was mainly affected at the beginning of the experiment, with higher mortality during the first three months in captivity (34.4%), probably because of the stress generated by routine management. After this period, mortality was lower and mainly associated with the spawning season; mortality rates of close to 10% were regularly recorded after maximum gonad maturation or marked gonadal development, while the mortality between the autumn–winter months of the four years was close to 0.0%. At the end of the experimental period, the survival of the wild population was 2.3%. 

In relation to growth, the present study demonstrates that *G. maculatus* increases in length throughout its lifespan (Figure 2C). The highest mean values of weight (3.52 ± 0.17 g) and total length (8.01 ± 0.42 cm) were recorded in the fourth year, with significant differences between years. The K-factor for this species in captivity increased significantly during the first year of the experiment, with maximum values between January and July. Then, this parameter decreased significantly, remaining stable for two years. In the fourth year of experiment, there was a new increase, but less than the first year (Figure 2D).

Figure 2E shows the instantaneous growth rate or gain in weight (GW) and length (GL) of the species. GL increased considerably until March, when an average of 4.12 ± 0.15 cm was recorded, while GW increased substantially until May (0.35 g). In the second and third year, the growth rate was noticeably lower than the first year, but with a similar trend that showed an increase during the autumn months (March to June) and a marked decrease during the spring and summer months. Between August and October of the first year, and in October of the following three years, the instantaneous growth rate registered negative values. 

The sex ratio did not differ from 1:1 (male:female), for either the wild population (48.9% and 51.1%, respectively) or the F1 (54.4% and 45.5%, respectively). 

### 3.2. Gonadosomatic Index (GSI)

The periods of lower temperature and hours of light (Figure 3A) coincided with the period of higher GSI in both female and male *G. maculatus*. GSI in males increased markedly from March (4.03 ± 0.20%), until reaching the maximum value (12.14 ± 0.74%) in August. Females maintained a GSI of close to 4.0% until June; and then increased rapidly, surpassing the males in July (10.24 ± 0.93%) and reaching the maximum value (17.7 ± 2.70%) in August. From August, GSI in both females and males began to markedly decrease, until registering values in January of 3.52 ± 0.93% and 3.8 ± 1.12%, respectively (Figure 3B). 

### 3.3. Sexual Maturity Stages

The wild population of *G. maculatus* reached first sexual maturity before spring (August–September) during their first year of life, with an average weight of 0.85 ± 0.01 g, a total length of 4.85 ± 0.16 cm, and a condition index of 0.0074. The progeny (F1) of the wild population reached first sexual maturity after one year of age. Thus, 69.04% of the F1 population reached first sexual maturity between August and September; prior to spring. 30.12% of this population were males at maximum maturity with a condition index of 0.0056, while 21.75% were females at maximum maturity with a condition index of 0.0072. Individuals that did not reach sexual maturity in the first year (identified as immature) had lower weight (0.60 ± 0.19 g), length (4.69 ± 1.10 cm), and condition factor (k = 0.0048). Initial maturity in males was recorded between April and June, and then from October to February, with a high percentage of fish in this state during December and January (65%) (Figure 4). Throughout the year, males were found in a state of advanced maturity, with higher percentages between May and July (62.0 ± 4.0%). Finally, maximum maturity occurred between April (5%) and December (7%), with a higher percentage of males between August and October (58 ± 6.0%). In regard to resorption, it was not possible to macroscopically differentiate the individuals in this state. 

Unlike the males, the females presented immaturity and initial maturity throughout the year. Additionally, females in advanced maturity were found during 11 months of the year, with a higher percentage in July (69%). Maximum maturity occurred between August and November, with a greater population of females in this state in August (73%) and September (54%). Resorption began in August with 3% of the population, and extended until November, a period in which 45% of immature females were found (Figure 4A). 

The GSI values for each stage of maturity are described in Table 1. Immature (virginal) individuals were analysed as a single group, as sex could not be differentiated macroscopically; therefore, the GSI values shown are the same for both groups. In general, females presented a higher GSI in all stages of maturity compared to males. 

### 3.4. Spawning Period and Frequency 

The population of *G. maculatus* held in captivity spawned between July 19 of the first year and February 7 of the following. During this period, two spawning subperiods were identified: one of greater intensity that extended throughout September and the first week of October; and another of lower intensity in the first half of summer. In the first subperiod, a total of 1978 embryos were obtained, while at the end of the second subperiod in February, only 38 embryos were obtained (Figure 5A). 

The individuals of *G. maculatus* caught in the river spawned spontaneously after the first year of age (between September and November). During this period, the females presented between 3 and 10 spawns; consequently, small groups of embryos were found in the substrate. The number of embryos per female and per spawning was highly variable. Thus, the minimum number of embryos collected per female in a day was 1, while the maximum was 429. The total number of embryos per female during the entire season evaluated varied between 163 and 1044, with an average of 467 ± 347 embryos. 

The spawning period recorded for the six females held in individual tanks coincided with that of the females held in groups. The average fecundity of these females was 91.2 ± 12.45% throughout the period, which indicates that the two males that shared the tank with each female had a much longer period of maximum maturity than the females. The above could also be corroborated in fish held in groups in other tanks, where it was observed that some males maintained fluid milt for much of the year (Figure 5B).

### 3.5. Fecundity

The batch fecundity recorded in the individually held females was highly and positively correlated (0.87) with the total weight (Figure 6). Thus, the mean value of batch fecundity varied between 382.6 and 1132.6 in females with the lowest (0.7–1.0 g) and highest body weight (1.9 and 2.2 g), respectively. Total length and batch fecundity presented a medium correlation (r = 0.4). 

## 4. Discussion

This study describes the main growth and reproductive parameters of a population of *G. maculatus* held under experimental conditions in southern Chile. Some specimens of the *G. maculatus* population recorded a lifespan of four years in captivity, while the rest of the population died during the experiment. It is important to highlight that the high mortality rate recorded during the first three months of the experiment can be explained by the process of adaptation to conditions of captivity which implied changes of habitat, culture density, feeding, and environmental factors, resulting ultimately in alterations of the physiological functions [26,27]. However, a small percentage (2.3%) of the population recorded a lifespan of four years. This coincides with the recent findings of Rojo and Boy [28] in Tierra del Fuego, Argentina, where otolith analysis was used to estimate an age of between 1 and 4.5 years for diadromous and landlocked *G. maculatus* populations. The present study also agrees with that described by Pollard [5], who estimated an age of three years using otolith analysis on a landlocked *G. maculatus* population in New Zealand. Together, these studies demonstrate that some individuals of this species can live up to 4.5 years. 

Taking into account initial mortality and mortality associated with the first reproductive event, the survival percentage in captivity after the first year of life was 55%. These results demonstrate that *G. maculaus* is an iteroparous species. This differs from what has been suggested by other authors who categorise the species as semelparous, either because individuals die following reproduction after the first year of life [24] or only a few individuals survive more than a year [1].

Mortality associated with reproduction in this study was two to five times lower than that recorded in the same species in a pioneering study on experimental cultivation [29]. Therefore, our results are positive, considering that we worked with undomesticated specimens. Normally, domestication processes, such as the one carried out in this study, involve the catching, transport, and acclimatisation of individuals to captive conditions. This, in turn, results in endocrine and physiological alterations, and immunosuppression [27,30]. Furthermore, post-reproduction mortality is associated with high energy and nutritional investment in gonadal growth, migration, spawning, and display and mating behaviours [31,32]. In fish from cold water, it is also related to low food availability and, therefore, low energy intake [33]. The mortality identified in this study was mainly influenced by (i) stress factors caused by routine management, such as cleaning, feeding, growth control practices, and sampling to determine the state of sexual maturity and (ii) marked gonadal development.

In relation to growth, the present study demonstrates that, like most fish, *G. maculatus* continues to grow throughout its life cycle, and at a higher rate in length and weight during the first six to seven months. These results also agree with those reported by Pollard [5] for wild populations using otolith techniques. However, the growth rate and length at end of life differ between studies. Pollard [5] reports length values of 9 cm in the first year, 14 cm in the second, and 17 cm in the third; in the present study, the fish reached a length of 4.12 cm after approximately six months, and only 8.01 ± 0.42 cm at the end of four years. In general, the mature adult individuals in this study had a smaller body size compared to those found in previous studies [1,2,15,25,34]. These differences could be related to several factors such as genetic variations influenced by the life history of the species [35,36,37], as well as differences in environmental conditions [28,38]. The differences between wild populations are probably associated with the availability of food, as well as the possibilities of migration. Migratory fish are able to find better food, which is reflected in better growth [39]. For example, morphological differences have been found between populations of wild fish that feed in pelagic and coastal areas [40]. The smaller size of the fish in this study compared to the size recorded in wild populations can be attributed to stress conditions due to handling, as well as the change in diet. Nutrients and diet composition are environmental factors that influence the somatic growth of teleost fish [41]. Most of the studies mentioned above used adult fish captured directly from the natural environment. Therefore, the diets of these fish are based on aquatic and terrestrial vertebrates to cover their nutritional requirements [38]. On the contrary, the fish in this study only received live food during the first phases of life. In the adult phase, salmonid food was used, which may not necessarily have covered the nutritional requirements of the individuals. Therefore, future studies are needed to investigate the nutritional requirements of this species during the different stages of development to ensure the sizes required for the market. 

In the present study, we found a low growth rate during autumn. This finding corroborates that described by Pollard [5], who assumed, based on observations of wild *G. maculatus* populations, that the growth rate of this species decreases during autumn. The decrease in growth rate in this study coincides with the maximum sexual maturity of the species and spawning. In fact, the instantaneous growth rate presented negative values during the reproductive peak. Such changes were also detected in the same species after sexual maturity by Mitchell [29]. Interestingly, Boy [14] found that *G. maculatus* experimentally exposed to “winter” conditions did not reduce their energy demands (as expected by temperature) and, although there were not changes in the basal metabolism during the experiment, they did identify lower gross conversion efficiency, a total absence of growth in length and weight, and a decrease in energy reserves. Furthermore, Chapman et al. [38] mentioned that diadromous populations that return to freshwater from the sea present a reduction in total length, without changes in head length. According to previous studies, these changes could be associated with a greater mobilisation of energy reserves towards reproduction. In particular, fish use energy reserves and calcium during gametogenesis, which, in some circumstances, affects the growth of body tissue [15,42].

The maximum GSI values recorded in this study, especially among males, are two to three times lower than those described in individuals caught in the wild in the extreme south of South America and a river in Otago, New Zealand. In females, the difference is less pronounced compared with previous studies, which have reported maximum values of between 22 and 26% [1,34]. According to Boy [34], the highest GSIs were obtained in individuals aged two and three years. Therefore, our hypothesis is that such differences are related to the age of the individuals, which is not only reflected in a larger body size, but also gonad size. Additionally, differences could also be influenced by a better and greater availability of food in wild conditions, as well as genetic variations. In temporal terms, these results do not completely agree with those found in the three studies mentioned above. In fact, Wansbrough [1] found that the GSI peak in the species differs between two rivers in the same region. Consequently, these results demonstrate that there are not only temporal differences, but also spatial differences in the growth and sexual maturity of *G. maculatus*, which is part of the phenotypic plasticity of the species. 

The present study demonstrates that *G. maculatus* reaches puberty in the first year of life. According to macroscopic analyses, about 50% of the individuals reached maximum maturity during the first year of age. The males underwent a marked gonad development from around six months (in March), while it occurred intensely in the females from June, with a GSI peak in both sexes in August. This indicates that females make a rapid and high reproductive effort that concludes with the first spawning period between September and November, at the first year of age. A second spawning period was also recorded between December and February. This supports the idea that *G. maculatus* can reach full sexual maturity in freshwater, and thus complete its entire life cycle in this environment [3,4].

Although macroscopic analyses only demonstrate maximum maturity between April and December for males and between September and November for females, the presence of embryos in the tanks between December and early February indicates that a small number of individuals presented maximum maturity in that period. The reason why these individuals were not identified could be because only 50 fish from a population of 567 individuals were macroscopically sampled in a single monthly sampling. According to Love [43], many fish become sexually mature when they reach a critical size, rather than a certain age. Consequently, this could explain why the fish of smaller size and condition index in the present study did not reach sexual maturity in the first year of life, with only those that were approximately 5.5 cm in length doing so. The embryos found in the second sub-period were probably the product of the spawning of fish that did not reach sexual maturity in the first subperiod.

In terms of length, the spawning period recorded in this study is in agreement with that registered in some populations occurring in Australia. Nevertheless, it seems that both maturity and spawning period vary among populations, even in the same geographic area. In landlocked populations, the spawning period varied between 6 to 7 months depending on the lake, while the shortest reproductive season was recorded in a hypersaline lake [6]. Similarly, the reproductive season of *G. maculatus* in Chile is similar to those populations distributed in Tierra del Fuego, Argentina [15]. This could be due the similarities in the season, and environmental conditions, especially photoperiod. The large reproductive season registered in this species is a great advantage compared with other species, even with those well established in the Chilean aquaculture, like coho salmon (*Oncorhynchus kisutch*), who show a short spawning period (May to June). In this regard, this species has potential to be included in culture and breeding programs, and therefore to diversify the Chilean aquaculture. 

With regard to the first hypothesis, the results of this study demonstrate that *G. maculatus* is able to spawn spontaneously in the tanks. That means that migration is not a prerequisite for the spawning of landlocked populations of this species. Furthermore, hormonal induction was not necessary as indicated for some migratory fish [44,45]. Probably, the spawning in this species is mainly mediated by environmental conditions, such as photoperiod, water temperature, and substrate. 

On the other hand, for several decades, there have been disagreements about whether or not *G. maculatus* is a semelparous species [15,24,34]. Based on the survival of some reproducers and the size frequency, Stevens et al. (2016) [25] proposed the iteroparity of the species. However, it is still unknown how many females actually manage to spawn again, under natural and captive conditions. Spawning monitoring carried out for 70 days on the six females showed that *G. maculatus* presents synchronous maturation in groups, which is in agreement with Mitchell [29]. Furthermore, the six females managed to survive after this period, showing iteroparous behaviour, while the post-reproduction mortality in the two main tanks was only around 10%. This results are in disagreement with our second hypothesis raised from several previous studies that estimated that the majority of individuals die after spawning [6,25]. On the contrary, this study demonstrates that most of the individuals of this species are able to survive after first reproduction, and have a lifespan greater of one year. Therefore, we now hypothesise that the semelparity pattern in this species can be characteristic of diadromous populations; consequently, future studies should be conducted in that population. 

The fecundity of females held in captivity in this study varied between 164 and 1578, depending on the weight. Firstly, it is important to highlight that the total fecundity in this species depends on the type of populations; consequently, diadromous populations show higher fecundity (up to 13,500 oocytes per female) than landlocked ones (up to 7400 oocytes per female) [15]. For this reason, and despite the different methodology used, the comparisons are only performed with landlocked populations. In general, our results are lower than those reported by other studies for wild females, but taking into account the type of spawning exhibited by this species and the methodology used in the different studies, it is likely that our results are within the range reported in previous studies. In fact, the batch fecundity of the heaviest females in this study is slighter lower than those of Fuegian *G. maculatus* in Argentina, where an average fecundity of 1422 ± 422 hydrated oocytes per ovary was described [15]. Another study in a landlocked population of the same species reported a fecundity of 107 to 2825 oocytes per female [46]. Recently, for the southernmost landlocked female populations, an average value of 1042 ± 598 was found with a range of between 162 and 3004 hydrated oocytes per ovary [34]. The highest fecundity in landlocked populations was informed in an early study in the Calle-Calle and Valdivia rivers in southern Chile, with a fecundity of between 390 and 7400 oocytes per female [3]. As pointed out above, this difference is mainly explained by the difference in the methodologies used to quantify the fecundity. In the present study, the fecundity was calculated as the total number of eggs collected by abdominal massage (batch fecundity) while in the aforementioned studies, it was quantified as the total number of oocytes present in whole formalin-fixed ripe ovaries (total fecundity). As this species exhibits synchronous maturation by groups, it is assumed that only mature oocytes were counted in this study. Secondly, the variation in fecundity could also be explained by the size and age of the females, where larger females present greater fecundity. Several studies have reported that the number of eggs produced by a female increases with size [3,47,48,49]. In the present study, a high correlation was found between fecundity and weight among females, but not length; thus, heavier females produced more eggs compared to lighter females. On the contrary, Boy et al. [15] reported no correlation between fecundity, weight, and length for this species. Fecundity can also be affected by several factors, including environmental sources, nutrition, genetic differences, or a combination of them [50,51,52,53]. 

On the other hand, the variable number of embryos found in the tanks can be explained by the fact that this species exhibits egg cannibalism. Similarly, to other galaxiids, *G. maculatus* prey on their own eggs immediately after spawning, when the eggs are still submerged. It could be deemed that captivity conditions facilitated the prolongation of this behaviour until the moment the embryos were removed from the tanks. Therefore, special care must be taken in future experiments, and especially when commercial exploitation will be considered. 

## 5. Conclusions

The present study demonstrates that *G. maculatus* can be cultured successfully in captivity without compromising the reproductive cycle; in fact, 50% of the population achieved its first reproductive event at one year of age, while the rest of the population progressively made the same, showing thus a spawning period of about seven months. The length of the spawning period is a positive factor for the use of this species in aquaculture, as well as in laboratory studies. Nevertheless, the growth, and therefore the body size, of the individuals was smaller than those found in previous studies in wild populations, which could limit or delay its inclusion in aquaculture. This information will be useful to continue the search for strategies for the reproductive management of the species in captivity and thus ensure high fertilisation rates, larvae, and fry production, as well as high-quality broodstock for future production cycles of the species. Finally, it is important to highlight that in reproductive and growth terms, our results could be affected by the origin of the population, diet, and stress factors. Therefore, further experiments should be carried out to determine similar parameters in the best-adapted populations and to assess the nutritional requirements of the species. 

## Figures and Tables

**Figure 1 animals-14-00320-f001:**
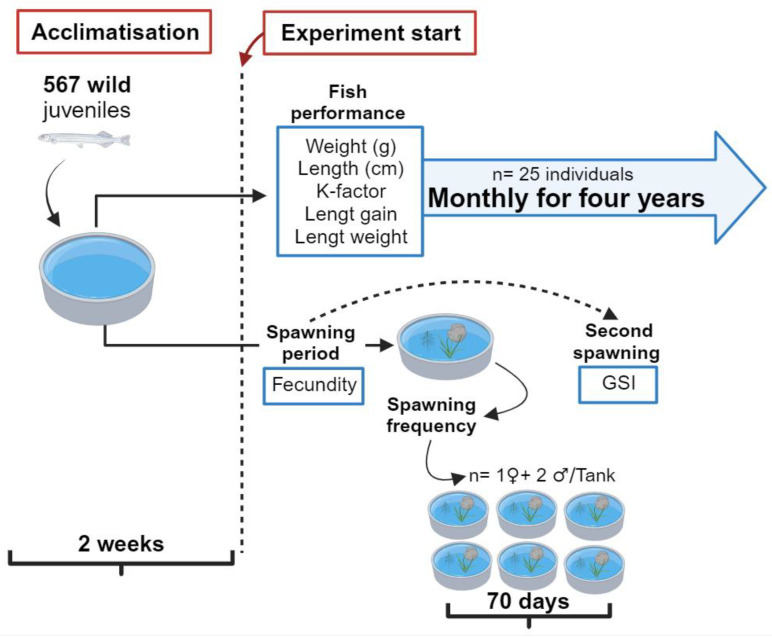
Diagram showing the main growth and reproductive parameters in an experimental culture of *G. maculatus* in southern Chile.

**Figure 2 animals-14-00320-f002:**
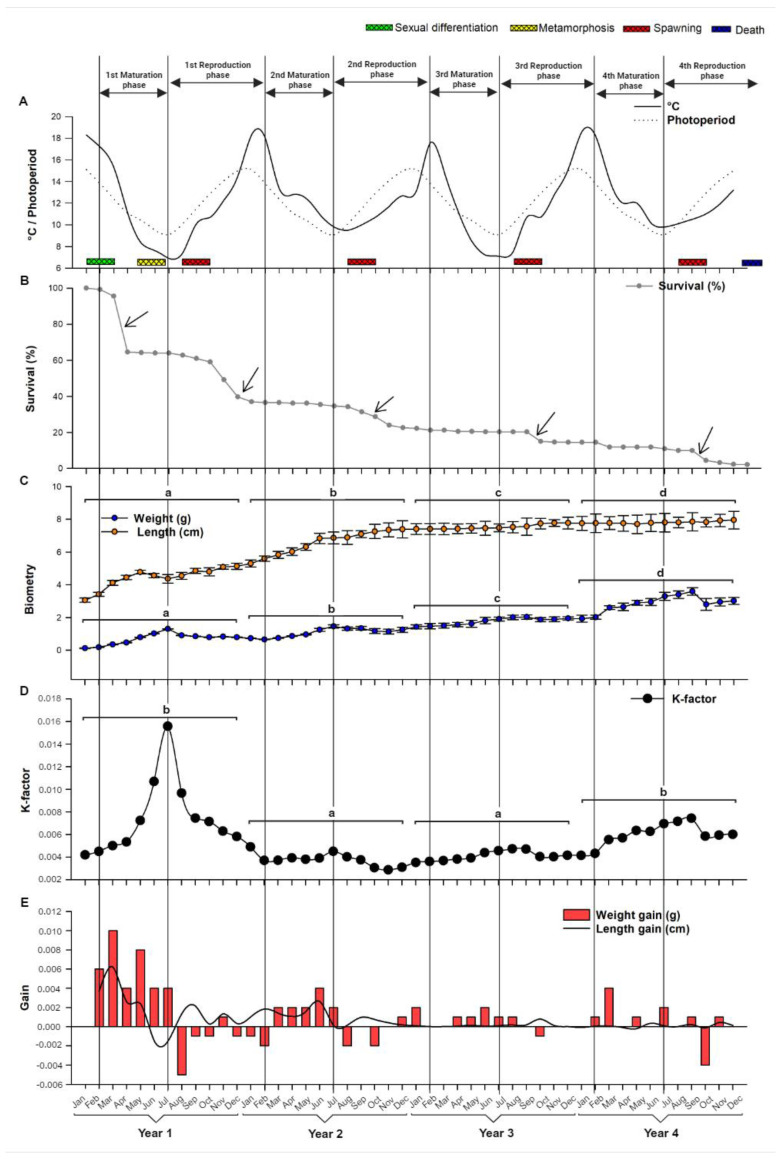
(**A**) Water temperature and natural photoperiod of an experimental culture of *G. maculatus* in southern Chile. (**B**–**E**) Results of survival and growth of *G. maculatus* throughout the experiment period. Solid and dashed lines in (**A**) indicate temperature and photoperiod, respectively. Arrows marked in (**B**) indicate periods of maximum mortality. The square brackets in (**C**,**D**) are used to indicate the years and the lower case letters above them indicate significant differences (*p* < 0.05) between the years.

**Figure 3 animals-14-00320-f003:**
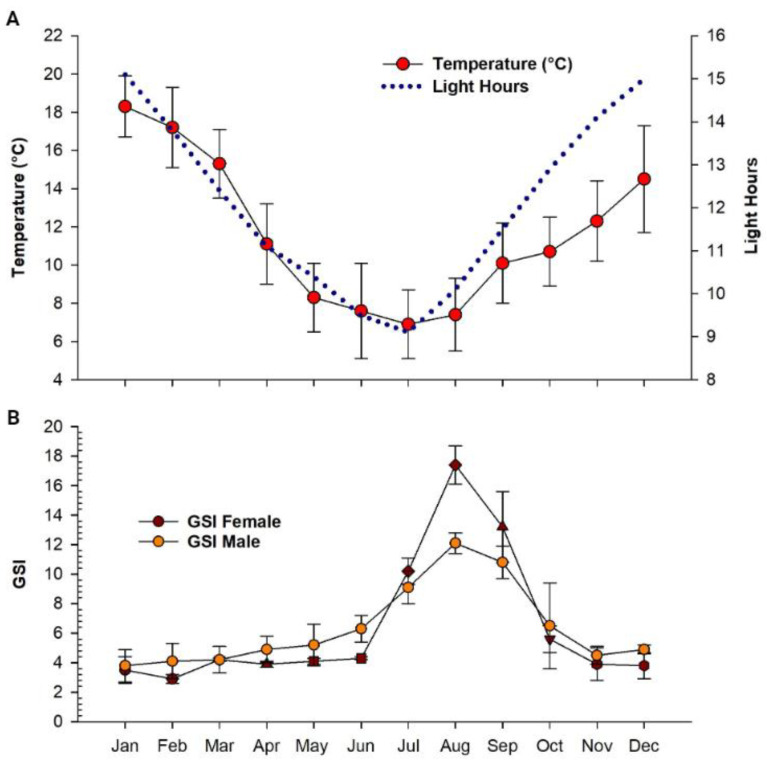
Monthly variations in GSI (mean ± SD), hours of light, and water temperature registered in an experimental culture of *G. maculatus* in southern Chile. (**A**) Hours of light and water temperature registered in an experimental culture of *G. maculatus* in southern Chile. (**B**) Monthly variations in GSI of males and females of *G. maculatus* measured through one year of experimental culture.

**Figure 4 animals-14-00320-f004:**
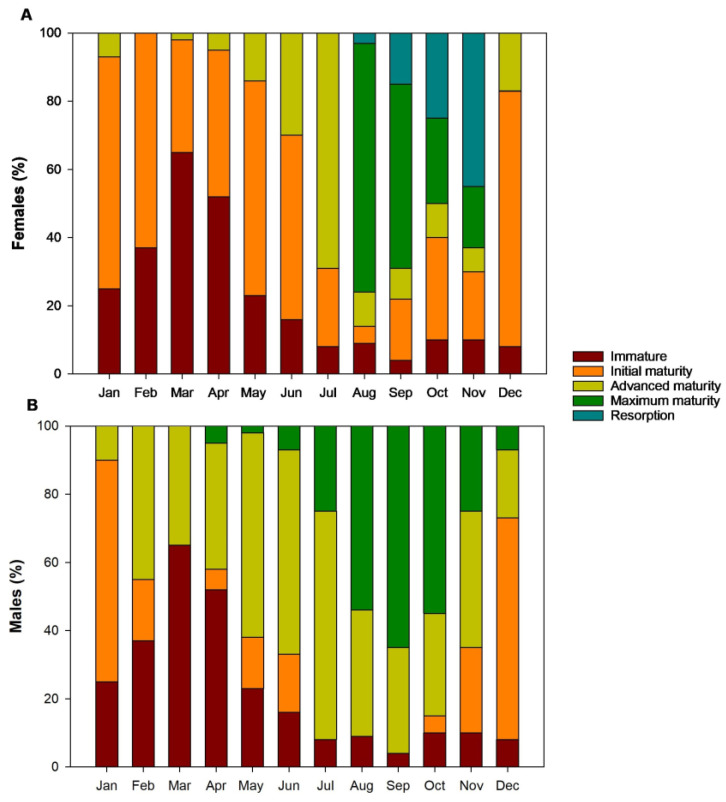
Percentage of (**A**) females and (**B**) males of *G. maculatus* in different stages of sexual maturity over a year under experimental culture conditions in southern Chile.

**Figure 5 animals-14-00320-f005:**
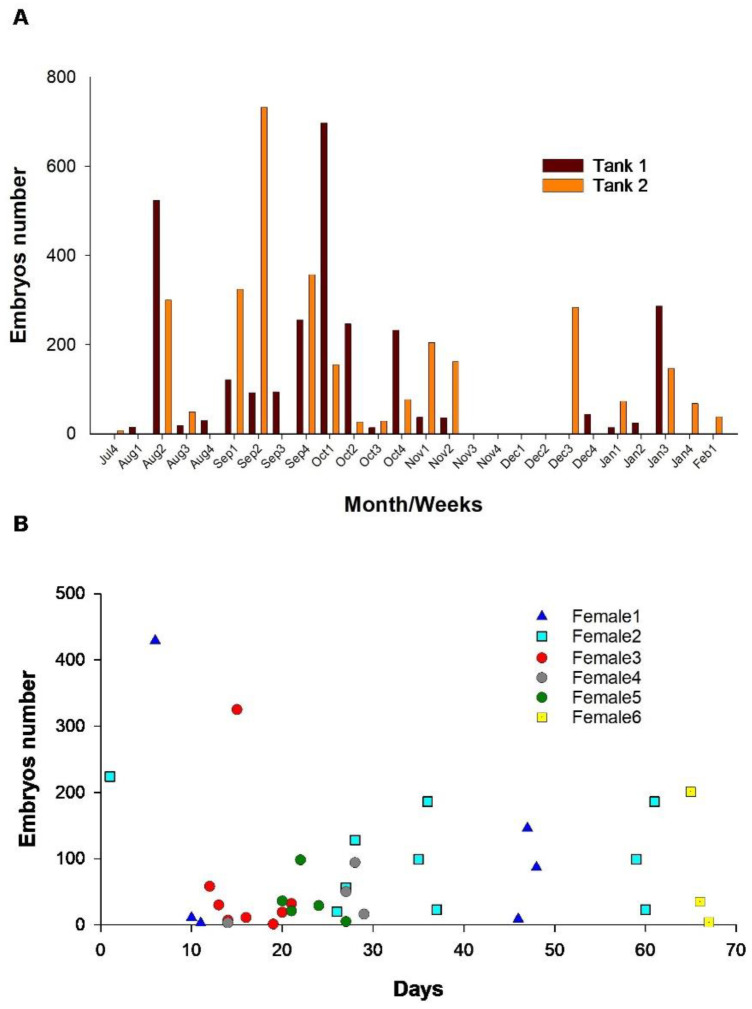
(**A**) Number of embryos collected per tank after spontaneous spawning of *G. maculatus* specimens held in captivity. The numbers that appear next to the month correspond to the week of that month. (**B**) Number of embryos collected per day after spontaneous spawning of females held in individual tanks.

**Figure 6 animals-14-00320-f006:**
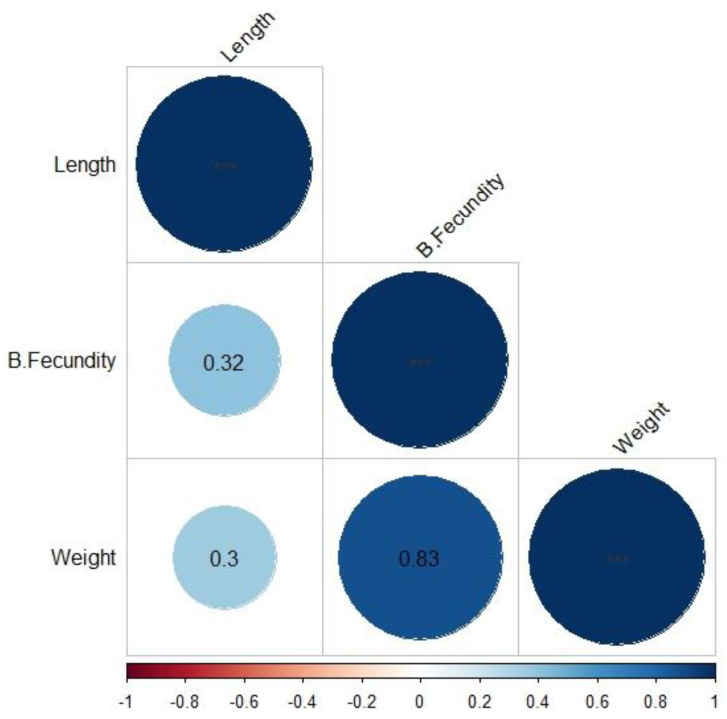
Correlation between the variables total weight (g), total length (cm), and batch fecundity (B.Fecundity) of female *G. maculatus*.

**Table 1 animals-14-00320-t001:** Gonadosomatic index (GSI) determined in male and female *G. maculatus* in different stages of sexual maturity.

Sexual Maturity Stage		GSI (%)						
	*n*	Males			*n*	Females		
		Mean ± SD	Minimum	Maximum		Mean ± SD	Minimum	Maximum
Virginal	50	0.91 ± 0.46	0.09	1.83	50	0.91 ± 0.46	0.09	1.83
Initial maturity	40	1.10 ± 1.62	1.12	5.42	50	4.00 ± 1.40	2.07	6.86
Advanced maturity	40	3.00 ± 1.03	3.21	7.23	30	9.90 ± 1.60	7.22	12.56
Maximum maturity	32	9.70 ± 2.12	4.22	10.56	46	17.70 ±2.92	13.03	25.16

Data are shown as mean ± SD.

## Data Availability

Data are available upon request from the authors.

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
