# Peer review of "The Reproductive Biology of Puye (Galaxias maculatus) under Experimental Culture"

_animals, 2024, doi:10.3390/ani14020320_

Round 1

Reviewer 1 Report

Comments and Suggestions for Authors

The article investigates the reproductive patterns of puye (Galaxias maculatus) under culture conditions. The study highlights the importance of understanding the reproductive biology of puye for sustainable aquaculture. The article suggests the need for further research to compare these findings with natural habitats and to address limitations in the study design. The results provide valuable insights into the reproductive patterns of puye in a controlled culture environment, which can contribute to the development of effective breeding strategies and management practices for puye aquaculture. However, some modifications need to be made before publication. Specific comments are attached as follows.

Abstract:

Ensure consistent use of terminology throughout the article. For example, use either "puye" or "Galaxias maculatus" consistently instead of alternating between the two.

Introduction:

Line 80: The introduction provides a brief overview of the puye species, its distribution, and its ecological and socioeconomic importance. However, it lacks a clear statement of the research objectives and the significance of studying the reproductive biology of puye under culture conditions. It would be helpful to include a clear research question or hypothesis and explain why understanding the reproductive patterns is important for sustainable aquaculture.

Line 43:Restructure the sentence: "At the end of the 1960s, the catch volume in Valdivia, Chile, varied between 1.5 and four tonnes per year; in 1990, fishing expanded to other regions and the volume reached 14 tonnes per year" for better clarity and readability.

Methods:

Line 89-100: Expand the description of the experimental setup, including details on the size and type of the culture system, water parameters (e.g., temperature, pH, dissolved oxygen), feeding regime, and any other relevant information.

Line 121: Provide more details regarding the measurements and observations conducted, such as the specific methods used for determining sexual maturity stages.

Discussion:

The discussion section should provide a comprehensive analysis and interpretation of the results in the context of the research objectives and existing literature. However, the current discussion is brief and lacks depth. To improve the quality of the discussion, please consider the following suggestions:

Compare the obtained results with previous studies on puye's reproductive biology, both in natural habitats and in captivity. Discuss any similarities or differences and explain their potential implications.

Provide a more detailed interpretation of the findings related to sexual maturity, spawning period, and fecundity. Discuss how these parameters may impact the feasibility and success of puye aquaculture.

Address any limitations or potential sources of bias in the study design and data collection. Discuss how these limitations may have influenced the results and suggest areas for future research.

Conclusion:

Line 465: Succinctly summarizes main outcomes. This part is too long.

Consider adding limitations and future research directions.

Comments on the Quality of English Language

as above

Author Response

Dear reviewer, 

Thank you for your comments and suggestion. All the corrections made to the manuscript are in attached. 

Reviewer 2 Report

Comments and Suggestions for Authors

Please refer to the note in the attached file and make corrections.

Comments on the Quality of English Language

Author Response

(The authors gave the same response as above.)

Round 2

Reviewer 2 Report

Comments and Suggestions for Authors

No additional comments.

Comments on the Quality of English Language

No additional comments.